# Persisting symptoms three to eight months after non-hospitalized COVID-19, a prospective cohort study

Arne Søraas[1]*, Karl Trygve Kalleberg[2], John Arne Dahl[1], Camilla Lund Søraas[3], Tor Åge Myklebust[4,5], Eyvind Axelsen[6], Andreas Lind[1], Roar Bævre-Jensen[7], Silje Bakken Jørgensen[8], Mette S. Istre[1], Eyrun F. Kjetland[1,9], Giske Ursin[5,10,11]

1 Department of Microbiology, Oslo University Hospital, Oslo, Norway, 2 Age Labs AS, Oslo, Norway, 3 Department of Environmental and Occupational Medicine, Oslo University Hospital, Oslo, Norway, 4 Department of Research and Innovation, Møre and Romsdal Hospital Trust, Ålesund, Norway, 5 Cancer Registry of Norway, Oslo, Norway, 6 Fürst Medical Laboratory, Oslo, Norway, 7 Department of Medical Microbiology, Vestre Viken Hospital Trust, Drammen, Norway, 8 Department of Clinical Microbiology and Infection Control, Akershus University Hospital, Lørenskog, Norway, 9 Nelson R Mandela School of Medicine, College of Health Sciences, University of KwaZulu-Natal, Durban, South Africa, 10 Dept of Nutrition, Institute of Basic Medical Sciences, University of Oslo, Oslo, Norway, 11 Dept. of Preventive Medicine, University of Southern California, Los Angeles, California, United States of America

* arne@meg.no

**Data Availability Statement:** De-identified data contains sensitive patient information and the data can only be shared after an approval process involving the Ethics committee and the Data

## Abstract

Long-COVID-19 is a proposed syndrome negatively affecting the health of COVID-19 patients. We present data on self-rated health three to eight months after laboratory confirmed COVID-19 disease compared to a control group of SARS-CoV-2 negative patients. We followed a cohort of 8786 non-hospitalized patients who were invited after SARS-CoV-2 testing between February 1 and April 15, 2020 (794 positive, 7229 negative). Participants answered online surveys at baseline and follow-up including questions on demographics, symptoms, risk factors for SARS-CoV-2, and self-rated health compared to one year ago. Determinants for a worsening of self-rated health as compared to one year ago among the SARS-CoV-2 positive group were analyzed using multivariate logistic regression and also compared to the population norm. The follow-up questionnaire was completed by 85% of the SARS-CoV-2 positive and 75% of the SARS-CoV-2 negative participants on average 132 days after the SARS-CoV-2 test. At follow-up, 36% of the SARS-CoV-2 positive participants rated their health "somewhat" or "much" worse than one year ago. In contrast, 18% of the SARS-CoV-2 negative participants reported a similar deterioration of health while the population norm is 12%. Sore throat and cough were more frequently reported by the control group at follow-up. Neither gender nor follow-up time was associated with the multivariate odds of worsening of self-reported health compared to one year ago. Age had an inverted-U formed association with a worsening of health while being fit and being a health professional were associated with lower multivariate odds. A significant proportion of non-hospitalized COVID-19 patients, regardless of age, have not returned to their usual health three to eight months after infection.

Protection Officer at our hospital. Both will follow the EU-GDPR rules. The Data Access Committee of the Norwegian Corona Cohort must be contacted through the PI: Arne Søraas, Department of Microbiology, Oslo University Hospital, Rikshospitalet, NO-0372 Oslo, Norway. The Norwegian Corona Cohort can be contacted through our e-mail address koronastudien@ous-hf.no.

**Funding:** AS and KTK have worked on the project paid from the company Age Labs. The funders had no role in study design, data collection and analysis, decision to publish, or preparation of the manuscript. The funder provided support in the form of salaries for authors AS and KTK, but did not have any additional role in the study design, data collection and analysis, decision to publish, or preparation of the manuscript. The specific roles of these authors are articulated in the 'author contributions' section.

**Competing interests:** Dr Søraas reported being an employee and shareholder at Age Labs outside of the submitted work. Dr Kalleberg reported receiving grants from the Norwegian Research Council during the conduct of the study and being a shareholder at Age Labs outside of the submitted work. This does not alter our adherence to PLOS ONE policies on sharing data and materials.

# Introduction

COVID-19 is a new disease caused by a beta-coronavirus, SARS-CoV-2. The virus has infected more than 165 million individuals and caused a pandemic which has killed more than 3.4 million people.

In Norway, a country with 5.4 million inhabitants and a free government-backed healthcare system, the first pandemic wave struck early in March 2020 as hundreds of Norwegians with COVID-19 returning from skiing holiday in the Austrian and Italian alps were not quarantined. A national lockdown was introduced March 12 and during the late spring and summer of 2020 the incidence of COVID-19 was low (<1/100.000 inhabitants per day) until the second wave began in September 2020.

Most COVID-19 patients have mild disease although a significant proportion experience severe disease and require hospitalization. Up to 61% report long-term complications including fatigue and memory problems after disease [1–6]. These symptoms have been proposed to constitute a syndrome termed "Post-Acute Sequelae of SARS-CoV-2 infection" (PASC) [7]. The mechanisms by which the virus causes short-term disease are not fully understood, but are thought to involve the direct action of the virus on different tissues including respiratory- and endothelial cells, but also the immune response directed towards the virus causes tissue damage [8]. The mechanisms behind PASC are even less understood, but damage caused by the virus and changes to the immune system after disease has been proposed [7, 9–11].

Knowledge of the prevalence and severity of PASC is important to inform decisions on population-wide infection control measures which are extremely expensive and to evaluate the long-term outcome of COVID-19 treatment both at home and in hospitals. Unfortunately, media reports on PASC may bias self-reported symptoms, obscuring the true incidence and also characterization of the syndrome.

We have addressed this important shortcoming by analyzing a questionnaire that was distributed to Norwegian COVID-19 patients and SARS-CoV-2 negative controls in the summer of 2020 before "long-covid" or PASC to our knowledge was reported in the Norwegian media. By using a validated questionnaire item with a known norm-value in the Norwegian population and a control group with a negative SARS-CoV-2 test, we investigated the self reported health compared to one year ago three to eight months after COVID-19.

# Materials and methods

The study was approved by the South-Eastern Norway Regional Committee for Medical and Health Research Ethics (REK 124170). Electronically signed written consent was obtained from all participants using the Norwegian National Identity number and two-factor authentication.

## Study design

This was a cohort study.

## Study population, ethics and measurements

Eligible participants were adults tested for SARS-CoV-2 with real-time RT PCR between February 1 and April 15, 2020, in four large accredited laboratories in South-Eastern Norway. Alle adults tested in this time period were invited by text messages or e-mail (S1 Fig in S1 File). Nearly all testing in Norway at that time was done on symptomatic patients and was free of cost [12]. In some periods, healthcare workers and patients with risk factors for severe COVID-19 were prioritized for testing. Willing participants signed an electronic consent form

and completed an online baseline questionnaire detailing demographics, preexisting conditions, symptoms, and risk factors for SARS-CoV-2 (S2 File). An online follow-up questionnaire also included a five-level single-item health transition question from the RAND-36 questionnaire comparing self-rated health to one year ago [13, 14]. Hospitalized patients were excluded from the cohort presented here. The study was approved by the Regional Research Ethics Committee according to the Declaration of Helsinki.

The outcome was self-reported health compared to one year ago at follow-up, and the exposure was the SARS-CoV-2 test result at baseline. SARS-CoV-2 status was obtained from laboratory records, while other data were based on the online questionnaires.

## Statistical analyses

All statistics was conducted using a two-tailed p-value of 0.05 as the significance level.

We conducted descriptive analyses using univariate logistic regression or the Pearson Chi-Square test to compare summary statistics or distributions for demographic variables, symptoms at baseline, and self-rated health compared to one year ago, between the exposure groups (SARS-CoV-2 positive or negative).

An adjusted logistic regression model was constructed using age and gender and the demographic variables with a statistically significantly different distribution between the exposure groups included as possible confounders.

The final multivariate logistic regression model included age (six groups), gender, history of chronic disease (history of cardiac disease, hypertension, chronic obstructive pulmonary disease, asthma, diabetes, cancer or on immunosuppressive medication or (yes/no)), smoking (never/ever), being a health professional, income level (four categories), self-reported fitness (three categories) and the time in days from testing to follow-up (continuous). The outcome variable was dichotomized into those reporting a worsening of health compared to one year or not (yes/no).

We conducted sensitivity analyses by subgrouping the SARS-CoV-2 positive participants on severity of disease (defined as the presence of self-reported fever, dyspnea, and fatigue on baseline), as well as analyses splitting the SARS-CoV-2 negative participants into a group with possible COVID-19 (i.e. reporting changes in the sense of smell or taste). We also analyzed each age group separately. Lastly, the final model was applied to each age group separately to explore the association between exposure group and the outcome (worsening of health) in each age group.

## Results

### Study population

A total of 2144 SARS-COV-2 positive and 31013 SARS-COV2-negative patients were invited. Of these, 853 (40%) SARS-CoV-2 positive and 8095 (26%) SARS-CoV-2 negative subjects agreed to participate. After excluding hospitalized patients we were left with 794 SARS-CoV2-positive and 7992 SARS-CoV-2 negative participants (Table 1). The final participants were, on average, 1.1 years younger than the invitees (p<0.001), and while 67% of the invitees were women, 73% of the participants were women (p<0.001). More than 95% of the participants were Caucasian.

### Baseline questionnaire

The baseline questionnaire was completed by all participants between March 27 and April 15, on average 16 days after the SARS-CoV-2 test (standard deviation (SD) 9 days).

**Table 1. Inclusion and retention.**

| Variable | SARS-CoV-2 status | |
| --- | --- | --- |
| | **Positive** | **Negative** |
| **Inclusion of participants** | | |
| **Eligible and invited** | | |
| Number invited, N | 2155 | 31013 |
| Females, % | 1056, 49% | 21089, 68% |
| Age, mean±SD | 49.6±17.4 | 45.9±16.6 |
| Consenting to participate and completed the baseline questionnaire, N | 853 | 8095 |
| Participation rate, % | 40% | 26% |
| Excluded from analysis, N[a] | 59 | 102 |
| Included for analysis, N | 794 | 7992 |
| Females, % | 429, 54% | 5994, 75% |
| Age, mean±SD | 47.3±13.9 | 44.8±13.2 |
| Days from testing to baseline questionnaire, mean±SD | 15±9 | 16±9 |
| **Three to eight month follow-up questionnaire** | | |
| Completed questionnaire, % | 672, 85% | 6006, 75% |
| Females, % | 382, 56.8% | 4520, 75.6% |
| Age, mean±SD | 48.5±13.5 | 46.1±13.0 |
| Days from testing to follow-up questionnaire, mean±SD | 126±33 | 133±36 |

[a] Participants that reported to be hospitalized.

SARS-CoV-2 positive participants were on average older (46% aged 50 or older vs 37% of SARS-CoV-2 negative), less likely to be health professionals, more likely to be male, less likely to report a chronic disease, less likely to be a smoker, more likely to be fit, and more likely to have high income than SARS-CoV-2 negative participants (all p-values 0.008 or lower) (Table 2).

Participants were asked to indicate symptoms they had experienced during the three weeks before completing the baseline questionnaire. For a majority of patients this period overlapped with their COVID-19 disease. The SARS-CoV-2 positive participants reported typical symptoms for COVID-19, including fever (66%), dyspnea (40%), cough (70%), a changed sense of smell and taste (69%) and fatigue (81%). Only 3% were asymptomatic, compared to 8% of the participants with a negative SARS-CoV-2 test (OR 0.3, p<0.001).

## Follow-up questionnaire

The follow-up questionnaire was completed by 85% of SARS-CoV-2 positive- and 75% of SARS-CoV-2 negative participants between July 1 and November 27, 2020 after up to five electronic reminders (Table 3). The average follow-up time was 132 days (SD 35 days) after testing. Respondents to follow-up were on average 1.3 years older than the baseline population (p<0.001), while there were no significant gender-differences between follow-up and baseline.

Symptoms reported significantly more frequently (p<0.01) at follow-up by formerly SARS-CoV-2 positive than negative participants were changes in senses of smell and taste (14% vs. 3%, OR 6.9, 95% CI 5.3–9.1), p<0.001), and "other symptoms" (8% vs. 2%, p<0.001). Several symptoms were reported more often by the SARS-CoV-2 negative group, including cough (8% vs. 17%, OR 0.4, p<0.001), body ache/muscular pain (8% vs. 12%, OR 0.7, 95% CI 0.5–0.9, p = 0.004), sore throat (7% vs. 22%, OR 0.3, 95% CI 0.2–0.4, p>0.001), nasal symptoms (11%

**Table 2. Study population and baseline status.**

| Variable | Level | SARS-CoV-2 positive (n = 794[a]) | | SARS-CoV-2 negative (n = 7992[a]) | | |
| | | N | % of positive | N | % of negative | P[b] |
| --- | --- | --- | --- | --- | --- | --- |
| Age group | 18–29 | 92 | 12% | 1085 | 14% | <0.001 |
| | 30–39 | 156 | 20% | 1959 | 25% | |
| | 40–49 | 178 | 22% | 2032 | 25% | |
| | 50–59 | 205 | 26% | 1678 | 21% | |
| | 60–69 | 122 | 15% | 950 | 12% | |
| | 70 and older | 41 | 5% | 288 | 4% | |
| Health professional | No | 553 | 70% | 2757 | 35% | <0.001 |
| | Yes | 234 | 30% | 5113 | 65% | |
| Gender | Female | 436 | 55% | 5944 | 75% | <0.001 |
| | Male | 353 | 45% | 2013 | 25% | |
| Chronic disease | No | 543 | 68% | 4848 | 61% | <0.001 |
| | Yes | 251 | 32% | 3144 | 39% | |
| Ever smoker | No | 487 | 62% | 4445 | 56% | 0.001 |
| | Yes | 294 | 38% | 3453 | 44% | |
| Fitness | In bad shape | 27 | 3% | 701 | 9% | <0.001 |
| | Fairly fit | 346 | 44% | 4523 | 57% | |
| | Fit | 421 | 53% | 2757 | 35% | |
| Household income | Below 300k | 14 | 2% | 154 | 3% | 0.008 |
| | 300k–600k | 101 | 15% | 1203 | 20% | |
| | 600k–1000k | 204 | 31% | 1642 | 28% | |
| | >1000k | 348 | 52% | 2895 | 49% | |
| **Symptoms the last three weeks before completing the baseline questionnaire, N (%)** | | | | | | |
| Fever | No | 274 | 35% | 5787 | 72% | <0.001 |
| | Yes | 520 | 66% | 2205 | 28% | |
| Temperature >39 degrees | No | 670 | 84% | 7746 | 97% | <0.001 |
| | Yes | 124 | 16% | 246 | 3% | |
| Dyspnea | No | 476 | 60% | 5711 | 72% | <0.001 |
| | Yes | 318 | 40% | 2281 | 29% | |
| Cough | No | 237 | 30% | 3701 | 46% | <0.001 |
| | Yes | 557 | 70% | 4291 | 54% | |
| Fatigue | No | 149 | 19% | 4443 | 56% | <0.001 |
| | Yes | 645 | 81% | 3549 | 44% | |
| Body ache, muscular pain | No | 326 | 41% | 5969 | 75% | <0.001 |
| | Yes | 468 | 59% | 2023 | 25% | |
| Sore throat | No | 432 | 54% | 3501 | 44% | <0.001 |
| | Yes | 362 | 46% | 4491 | 56% | |
| Changed sense of smell or taste | No | 248 | 31% | 7225 | 90% | <0.001 |
| | Yes | 546 | 69% | 767 | 10% | |
| Nasal symptoms | No | 428 | 54% | 4182 | 52% | 0.396 |
| | Yes | 366 | 46% | 3810 | 48% | |
| Headache | No | 259 | 33% | 4018 | 50% | <0.001 |
| | Yes | 535 | 67% | 3974 | 50% | |
| Abdominal pain, nausea or diarrhea | No | 513 | 65% | 6302 | 79% | <0.001 |
| | Yes | 281 | 35% | 1690 | 21% | |
| Other symptoms | No | 666 | 84% | 7488 | 94% | <0.001 |
| | Yes | 128 | 16% | 504 | 6% | |

(*Continued*)

**Table 2.** (Continued)

| | | SARS-CoV-2 positive (n = 794[a]) | | SARS-CoV-2 negative (n = 7992[a]) | | |
|---|---|---|---|---|---|---|
| Variable | Level | N | % of positive | N | % of negative | P[b] |
| Asymptomatic | Symptomatic | 774 | 98% | 7319 | 92% | <0.001 |
| | Asymptomatic | 20 | 3% | 673 | 8% | |

[a] Number missing for each variable: Health professional: 129, Gender 40, Smoking 107, Fitness 11, Household Income 2225. Symptoms were checkboxes without a "no" alternative.

[b] p-value for unadjusted logistic regression comparing those without the trait with those with the trait. For multi-level traits, the p value was calculated using Pearsons Chi Squqare.

Ref: Reference category.

vs. 24%, OR 0.4, 95% CI 0.3–0.5, p<0.001), headache (13% vs. 23%, OR 0.5, 95% CI 0.4–0.6) and abdominal symptoms (6% vs. 11%, OR 0.5, 95% CI 0.3–0.7, p<0.001).

At the follow-up questionnaire participants were asked to rate their health compared to one year ago. A total of 36% of the SARS-CoV-2 positive respondents rated their health "somewhat" or "much" worse than one year ago (Fig 1). In contrast, 18% of the SARS-CoV-2 negative participants reported a similar deterioration of health. The average in the Norwegian population in a non-pandemic period was 12.3% [15].

After adjusting for age, gender, chronic diseases, smoking, being a health professional, income level, fitness, and time from the SARS-CoV-2 test to follow-up, SARS-CoV-2 positivity at baseline remained strongly associated with a worsening of self-rated health at follow-up (OR 2.9, 95% CI 2.4–3.5, p<0.001) (Table 4).

When the SARS-CoV-2 positive participants were stratified by disease severity, those who reported to have a more severe disease (fever, dyspnea, **and** fatigue at baseline) were more likely to report a worsening of health at three months (OR 1.7, 95% CI 1.1–2.5, p = 0.009) compared to one year ago than those with mild symptoms (Fig 2).

In total, 767 SARS-CoV-2 negative participants (10%) reported a changed sense of smell and taste at baseline, consistent with a possible false-negative SARS-CoV-2 test. Of these, 28% reported a worsening of their health compared to one year ago (Fig 3). In the fully adjusted multivariate regression model, being in this group was also associated with a worsening of health compared to the SARS-CoV-2 negative group without changes in smell and taste (OR 1.7, 95% CI 1.4–2.1, p<0.001).

Finally, to investigate whether the association between COVID-19 and self-reported worsening of health was influenced by age, we ran the adjusted model on each age group separately and found that COVID-19 patients aged 30–49 had the highest odds ratio for self-reported worsening of health (> 4.0) while the COVID-19 patients from other age groups had an odds ratio lower than 3.0 compared to the COVID-19 negative patients. Worryingly, the odds ratio in the 18–29 age group was 2.5 (95% CI 1.3–4.7, p 0.007).

## Discussion

We aimed to measure whether non-hospitalized COVID-19 patients reported a deterioration of health compared to one year ago even before "long-COVID" or PASC was reported in the media and found that 36% reported a deterioration three to eight months after disease compared to 18% of controls. The normal value in a representative Norwegian population for this questionnaire item is 12.3% (11).

**Table 3. Follow-up symptoms the last three weeks before completing the three-month follow-up questionnaire.**

| Variable | Level | SARS-CoV-2 positive (n = 676 [a]) | | SARS-CoV-2 negative (n = 6006 [a]) | | P[b] |
|---|---|---|---|---|---|---|
| | | N | % of positive | N | % of negative | |
| Fever | No | 659 | 98% | 5748 | 96% | 0.029 |
| | Yes | 17 | 3% | 258 | 4% | |
| Temperature >39 degrees | No | 673 | 100% | 5967 | 99% | 0.524 |
| | Yes | 3 | 0% | 39 | 1% | |
| Dyspnea | No | 608 | 90% | 5525 | 92% | 0.066 |
| | Yes | 68 | 10% | 481 | 8% | |
| Cough | No | 623 | 92% | 5015 | 84% | <0.001 |
| | Yes | 53 | 8% | 991 | 17% | |
| Fatigue | No | 519 | 77% | 4801 | 80% | 0.053 |
| | Yes | 157 | 23% | 1205 | 20% | |
| Body ache, muscular pain | No | 623 | 92% | 5311 | 88% | 0.004 |
| | Yes | 53 | 8% | 695 | 12% | |
| Sore throat | No | 628 | 93% | 4711 | 78% | <0.001 |
| | Yes | 48 | 7% | 1295 | 22% | |
| Changed sense of smell or taste | No | 580 | 86% | 5866 | 98% | <0.001 |
| | Yes | 96 | 14% | 140 | 2% | |
| Nasal symptoms | No | 601 | 89% | 4575 | 76% | <0.001 |
| | Yes | 75 | 11% | 1431 | 24% | |
| Headache | No | 590 | 87% | 4602 | 77% | <0.001 |
| | Yes | 86 | 13% | 1404 | 23% | |
| Abdominal pain, nausea or diarrhoea | No | 637 | 94% | 5333 | 89% | <0.001 |
| | Yes | 39 | 6% | 673 | 11% | |
| Other symptoms | No | 622 | 92% | 5897 | 98% | <0.001 |
| | Yes | 54 | 8% | 109 | 2% | |
| Asymptomatic | Symptomatic | 380 | 56% | 3417 | 57% | 0.735 |
| | Asymptomatic | 296 | 44% | 2589 | 43% | |
| Self-assessed change in health last 1 year (summer) | A lot worse | 38 | 6% | 130 | 2% | <0.001 |
| | Somewhat worse | 205 | 30% | 933 | 16% | |
| | About the same | 386 | 57% | 4078 | 68% | |
| | A bit better | 32 | 5% | 617 | 10% | |
| | A lot better | 14 | 2% | 242 | 4% | |

[a] Number missing for Self-assesed change in health last 1 year (summer) was 12.

[b] p-value for unadjusted logistic regression comparing those without the trait with those with the trait. For multi-level traits, the reference level lowest level is reference (ref). For variables with several levels the odds ratio (OR) is for each level.

Our prospective study was conducted during the first wave in home-treated patients that received advice on using antipyretics and to contact a doctor in the case of severe symptoms including difficulty breathing and indicates that the medium-term outcome of this approach was not optimal.

The single-item health transition questionnaire in our questionnaire asked the respondents to compare their current health with that of one year ago. Other studies have shown that current health-related quality of life heavily influences the answers to this item which is closely related to the PASC syndrome [13, 14]. Thus, this question should identify patients experiencing a worsening of their health after COVID-19 with a high sensitivity, although many aspects

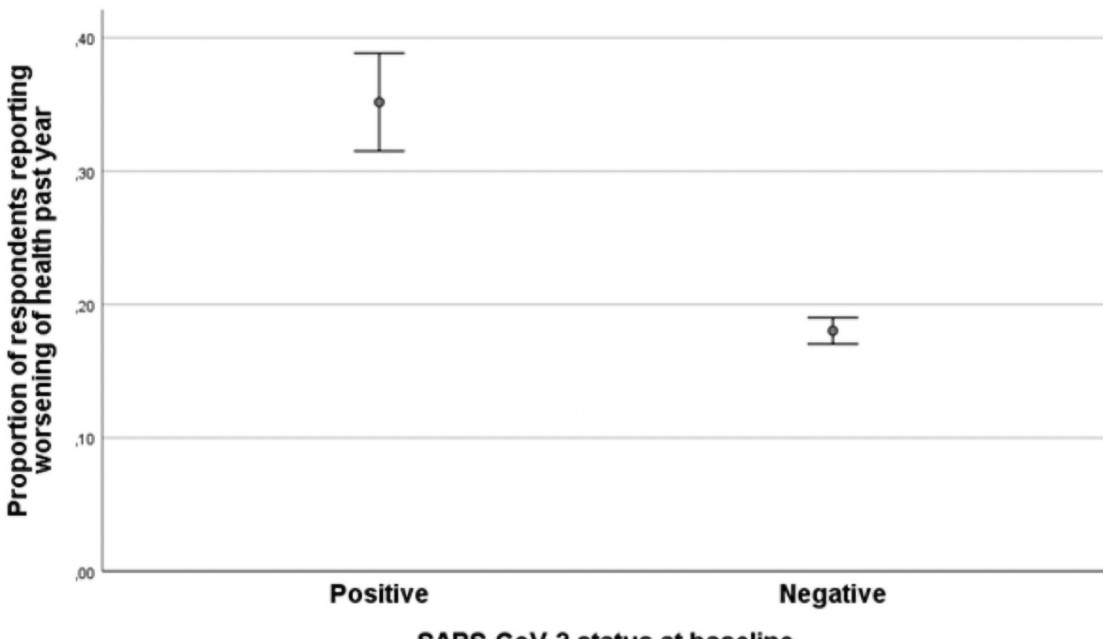

**Fig 1. Proportion of participants reporting a worsening of health past year at 3–8 months after baseline.** The figure shows the proportion of participants reporting a "somewhat" or "much" worse health than one year ago at the time of follow-up. SARS-CoV-2 positive or negative status was based on RT-PCR at Norwegian accredited laboratories at baseline. Error bars are 95% confidence intervals for the means.

of the quality of life after COVID-19 and why the health-related quality of life has changed will not be captured.

There were several significant differences in demographic variables between the exposure groups (Table 2). These can partly be explained by testing policy early in the first pandemic wave in Norway, where healthcare workers were prioritized and therefore overrepresented in the negative group. In Norway the majority of healthcare workers are female. Moreover, as in other countries, men are slightly overrepresented among COVID-19 positive patients compared to women which can also explain the higher proportion of women in the COVID-negative group.

At baseline, SARS-CoV-2 positive participants were more likely to have reported being "very fit" than those who tested negative. This is likely because the initial Norwegian cases in February/March 2020 were healthy ski tourists returning from the European Alps. However, being fit was itself found to be protective against a worsening of health, which is as expected given the multiple positive health effects of being fit.

At follow-up, airway symptoms were more prevalent in the SARS-CoV-2 negative group. Such symptoms were a prerequisite for being tested, and individuals frequently experiencing such symptoms are therefore expected to be tested more often even if they did not have COVID-19 and therefore the negative group will be enriched with such individuals. In itself, the low frequency of airway symptoms is expected to lead to a better health in the SARS-CoV-2 positive group; yet, the opposite was seen.

In the stratified analyses, we found that people of middle age were at a higher risk of PASC compared to the younger or older age groups with an adjusted OR of 4 compared to 2–3 for those younger than 30 or older than 50, but even young non-hospitalized patients were not protected from the COVID-19s negative long-term effects. Also, long-term deterioration of

**Table 4. Fully adjusted multivariable model for the whole population, comparing participants reporting a worsening of health compared to one year ago (yes/no).**

| Variable | P[a] | OR[b] | 95% CI |
|---|---|---|---|
| Age group 18–29 | | ref | |
| Age group 30–39 | 0.26 | 0.86 | 0.7–1.1 |
| Age group 40–49 | 0.87 | 0.98 | 0.8–1.3 |
| Age group 50–59 | 0.99 | 1.00 | 0.8–1.3 |
| Age group 60–69 | 0.81 | 0.97 | 0.7–1.3 |
| Age group 70-> | 0.10 | 1.35 | 0.9–1.9 |
| Gender (male) | 0.81 | 0.98 | 0.8–1.1 |
| Chronic disease (yes) | <0.001 | 1.75 | 1.5–2 |
| Ever smoker (yes) | 0.07 | 1.13 | 1–1.3 |
| Health professional (yes) | <0.001 | 0.69 | 0.6–0.8 |
| Household income[c] | <0.001 | 0.83 | 0.8–0.9 |
| Fitness | | | 0–0 |
| In bad shape | | ref | 0–0 |
| Fairly fit | <0.0001 | 0.53 | 0.4–0.6 |
| Fit | <0.001 | 0.35 | 0.3–0.4 |
| Days from baseline to F2 form completed date[d] | 0.08 | 1.00 | 0.997–1.0 |
| SARS-CoV-2 positive at baseline | <0.001 | 2.89 | 2.4–3.5 |

[a] p-value for the fully adjusted logistic regression model.

[b] OR (odds ratio) is for experiencing a worsening of health compared to one year ago compared to no health worsening.

[c] OR for household income is per income level.

[d] OR per day.

Ref: Reference category.

health was only weakly attenuated by time. However, most participants submitted their follow-up questionnaire during the summer limiting the study's power to detect attenuation over time.

Our results are consistent with a recent report of non-hospitalized COVID-19 patients, where 55% had at least one persisting symptom at six months follow-up, and persisting symptom severity was dependent on disease severity [6]. Our main finding, that a significant proportion of COVID-19 patients do not recover completely after three to eight months is also consistent with unpublished surveys without control groups in non-hospitalized and hospitalized patients [5, 16, 17]. These surveys also describe a possible long-COVID syndrome that include cognitive symptoms not covered in our questionnaires, which may serve to explain why our COVID-19 patients reported a worsening of health while their burden of airway symptoms at follow-up was lower than the controls. The pathophysiology of PASC is starting to be mapped out and includes vasculitis caused by endothelial damage, an inflammatory post-infectious syndrome and also possibly social isolation [7, 8, 10, 11]. Our results cannot differentiate between these hypotheses.

## Limitations

A limitation of the study is that knowledge of COVID-19 status at baseline could have led to participation or response bias in the follow-up round. However, although possible, we find it unlikely to have affected our results to a significant degree, and the lack of literature and

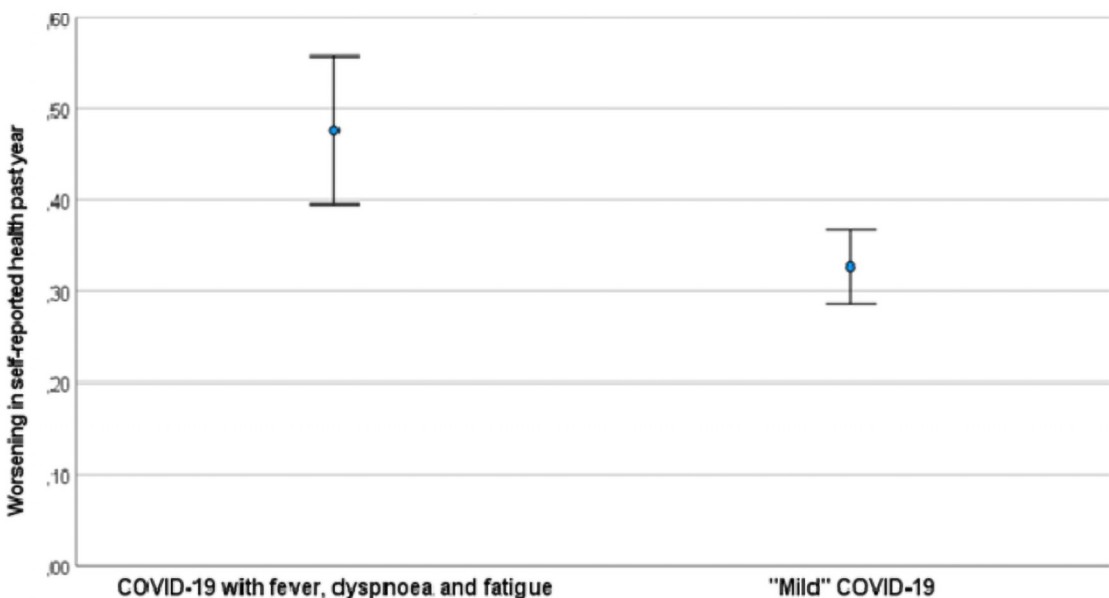

**Fig 2. Proportion of participants reporting a worsening of health past year at 3–8 months after baseline stratified on disease severity.** The figure shows the proportion of participants reporting a "somewhat" or "much" worse health than one year ago at the time of follow-up. Questionnaire data on symptoms during disease was used to classify patients into the two severity groups. SARS-CoV-2 positive or negative status was based on RT-PCR at Norwegian accredited laboratories at baseline. Error bars are 95% confidence intervals for the means.

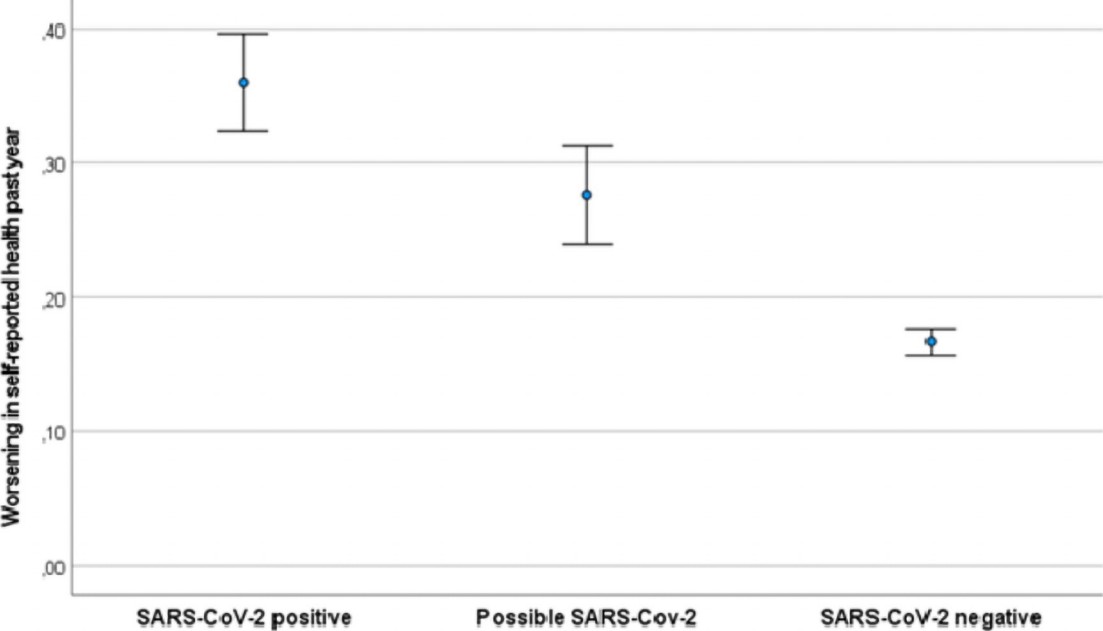

**Fig 3. Proportion of participants reporting a worsening of health past year at 3–8 months after baseline stratified on likelihood of previous COVID-19.** The figure shows the proportion of participants reporting a "somewhat" or "much" worse health than one year ago at the time of follow-up. Questionnaire data on symptoms at baseline was used to classify patients. "SARS-CoV-2 positive" status was based on RT-PCR at Norwegian accredited laboratories. "Possible SARS-CoV-2" was patients with a negative PCR test, but reporting changes to the senses of smell or taste at baseline. "SARS-CoV-2 negative" was PCR negative patients not reporting changes to the senses of smell or taste at baseline. Error bars are 95% confidence intervals for the means.

reports in the Norwegian media on the subject at the time the majority of follow-up questionnaires were completed in July 2020 is a major strength of the study.

We did not follow the participants with repeated PCR tests or serology and infections (or reinfections which have been demonstrated between baseline and follow-up may influence our results slightly, but the incidence of COVID-19 during this period was very low in Norway (0.24% between May and November 2020) [18]. Moreover, we did not collect information about airway symptoms among the participants before inclusion or information about other possible causes for airway symptoms at follow-up. Thus some of the symptoms reported after COVID-19 may have also have been present before the disease and may therefore have other causes than COVID-19.

Furthermore, it is likely that patients with very mild or severe symptoms during and after disease are underrepresented in the study as individuals with mild disease could be less interested in the subject, while hospitalized patients were excluded and other patients with severe symptoms may not have been able to participate.

Although we ran multivariate models adjusting for several likely confounders, there may still have been unmeasured or residual confounding in the association observed between SARS-CoV-2 status and subjective health at follow-up.

Finally, the results are not directly generalizable to the population as a whole because it was an electronic survey and older individuals, disadvantaged groups and individuals of other languages may be underrepresented.

A strength of the study is the prospective design where patients were recruited shortly after their disease and followed for up to eight months.

## Conclusion

We conclude that a significant proportion of non-hospitalized COVID-19 patients, regardless of age, gender and follow-up time, reported a worsening of health three to eight months after infection when PASC was to our knowledge not reported in the Norwegian media. In our cohort, this deterioration was not caused by airway symptoms, indicating that other manifestations of COVID-19 may be more important for the PASC syndrome.

Our results indicate that "mild" COVID-19 is less common than initially believed and provides a strong impetus for population-wide infection control and vaccination. The medium-term outcome of home-treated COVID-19 was not fully satisfactory.

## Supporting information

**S1 File. Inclusion flow diagram.** Flow diagram of the inclusion of participants in the Norwegian Corona Cohort.
(PDF)

**S2 File. Questionnaires.** English translation of relevant parts of the questionnaires used in the study.
(PDF)

## Author Contributions

**Conceptualization:** Arne Søraas, Camilla Lund Søraas.

**Data curation:** Arne Søraas, Karl Trygve Kalleberg, Mette S. Istre, Eyrun F. Kjetland.

**Formal analysis:** Arne Søraas, Karl Trygve Kalleberg, Camilla Lund Søraas, Tor Åge Myklebust, Giske Ursin.

**Funding acquisition:** Arne Søraas, John Arne Dahl.

**Investigation:** Arne Søraas, Karl Trygve Kalleberg, Eyvind Axelsen, Andreas Lind, Roar Bævre-Jensen, Silje Bakken Jørgensen, Mette S. Istre, Giske Ursin.

**Methodology:** Arne Søraas, Karl Trygve Kalleberg, John Arne Dahl, Camilla Lund Søraas, Tor Åge Myklebust, Mette S. Istre, Eyrun F. Kjetland, Giske Ursin.

**Project administration:** Arne Søraas, John Arne Dahl, Mette S. Istre.

**Resources:** Arne Søraas, Karl Trygve Kalleberg, Mette S. Istre.

**Software:** Karl Trygve Kalleberg.

**Supervision:** Arne Søraas, Karl Trygve Kalleberg.

**Writing – original draft:** Arne Søraas, Karl Trygve Kalleberg, John Arne Dahl, Camilla Lund Søraas, Eyrun F. Kjetland, Giske Ursin.

**Writing – review & editing:** Arne Søraas, Karl Trygve Kalleberg, John Arne Dahl, Camilla Lund Søraas, Tor Åge Myklebust, Eyvind Axelsen, Andreas Lind, Roar Bævre-Jensen, Silje Bakken Jørgensen, Mette S. Istre, Eyrun F. Kjetland, Giske Ursin.

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
