## [Decision Letter · Decision Letter 0]

23 Apr 2021

PONE-D-21-09363

Persisting Symptoms Three to Eight Months after Non-Hospitalized COVID-19, a Prospective Cohort Study in 8786 participants

PLOS ONE

Dear Dr. Søraas,

Your manuscript has been reviewed by three experts and their comments follow. They have made very specific suggestions and ask that you update your analysis and references. Please respond to their comments and return the paper as early as possible.

We look forward to receiving your revised manuscript.

Kind regards,

Dong-Yan Jin

Academic Editor

PLOS ONE

Additional Editor Comments:

See above.

Journal Requirements:

2. Please include additional information regarding the survey or questionnaire used in the study and ensure that you have provided sufficient details that others could replicate the analyses. For instance, if you developed a questionnaire as part of this study and it is not under a copyright more restrictive than CC-BY, please include a copy, in both the original language and English, as Supporting Information.  If the original language is written in non-Latin characters, for example Amharic, Chinese, or Korean, please use a file format that ensures these characters are visible.

3. Please state whether you validated the questionnaire prior to testing on study participants. Please provide details regarding the validation group within the methods section.

4. Thank you for providing the following Funding Statement: 

"AS and KTK have worked on the project paid from the company Age Labs.

The funders had no role in study design, data collection and analysis, decision to

publish, or preparation of the manuscript."

We note that one or more of the authors is affiliated with the funding organization, indicating the funder may have had some role in the design, data collection, analysis or preparation of your manuscript for publication; in other words, the funder played an indirect role through the participation of the co-authors.

If the funding organization did not play a role in the study design, data collection and analysis, decision to publish, or preparation of the manuscript and only provided financial support in the form of authors' salaries and/or research materials, please review your statements relating to the author contributions, and ensure you have specifically and accurately indicated the role(s) that these authors had in your study in the Author Contributions section of the online submission form. Please make any necessary amendments directly within this section of the online submission form.  Please also update your Funding Statement to include the following statement: “The funder provided support in the form of salaries for authors [insert relevant initials], but did not have any additional role in the study design, data collection and analysis, decision to publish, or preparation of the manuscript. The specific roles of these authors are articulated in the ‘author contributions’ section.”

If the funding organization did have an additional role, please state and explain that role within your Funding Statement.

Please also provide an updated Competing Interests Statement declaring this commercial affiliation along with any other relevant declarations relating to employment, consultancy, patents, products in development, or marketed products, etc.  

Reviewers' comments:

Reviewer's Responses to Questions

**Comments to the Author**

1. Is the manuscript technically sound, and do the data support the conclusions?

Reviewer #1: Yes

Reviewer #2: Yes

Reviewer #3: Partly

2. Has the statistical analysis been performed appropriately and rigorously? 

Reviewer #1: Yes

Reviewer #2: Yes

Reviewer #3: Yes

3. Have the authors made all data underlying the findings in their manuscript fully available?

Reviewer #1: No

Reviewer #2: No

Reviewer #3: Yes

4. Is the manuscript presented in an intelligible fashion and written in standard English?

Reviewer #1: Yes

Reviewer #2: Yes

Reviewer #3: No

5. Review Comments to the Author

Reviewer #1: I suggest to review recent literature data on the topic

e.g.

Sudre CH, Murray B, Varsavsky T, Graham MS, Penfold RS, Bowyer RC, et al. Attributes and predictors of Long-COVID: analysis of COVID cases and their symptoms collected by the Covid Symptoms Study App. medRxiv. Cold Spring Harbor Laboratory Press; 2020;2020.10.19.20214494.

and perform similar analysis to confirm results

Reviewer #2: Review: Persisting Symptoms Three to Eight Months after Non-Hospitalized COVID-19, a Prospective Cohort Study in 8786 participants

• Overall, I think that the question (if non-hospitalised adults continue to experience symptoms from PASC) is important and not fully answered in current literature

• The authors’ methodology to answer this question is to use a survey to compare self-reported health outcomes between COVID positive and negative patients at an adequate time post-infection; the use of a control group is an important positive

• However, the discussion is very limited and does not fully explain the research findings

Abstract

• I think it would be important to mention the numbers of COVID-positive and negative patients within the abstract, given most of the 8786 participants are COVID-negative

• While the abstract mentions the most important conclusion, it does not mention the other important data e.g. important symptoms such as cough are more common within the COVID-negative group

Introduction

• The introduction is limited and the authors have not discussed the current literature available regarding post covid symptoms following hospitalisation

• Having 8 references for the first sentence seems excessive, given the first five are not further referenced later

• Saying that ‘outpatients with COVID-19 experience relatively benign symptoms’ may trivialise what are often very debilitating symptoms and the authors should rephrase this.

• Why have the authors picked 3-8 months – was this an a priori decision?

• The introduction should clarify that it is the non-hospitalised COVID-19 patient group which is being assessed

Materials and methods

• Overall, these are described well

• I think the authors should clarify clearly how they classified severity of disease within a non-hospitalised cohort and the evidence used for this classification. I have normally seen ‘mild COVID’ used to describe non-hospitalised COVID.

• It would be useful to understand why subjects underwent a swab e.g. contact with COVID positive person, self referral with minimum symptoms, routine referral with no symptoms, ie were the negative patients likely to have negative swabs

Results

• Overall the results can be understood, although some of the tables could be simplified

• In the baseline questionnaire, COVID-negative patients were more likely to experience sore throat than COVID-positive patients. As this is the only symptom they were more likely to experience, I think it should be mentioned within the results section

• Can the authors add a sentence about household income to clarify if it is increased or decreased household income which is associated with a worsening of health?

• The most important finding that the authors have focussed on is ‘self-assessed change in health in last one year’. I think it is worth clarifying that while the authors administered two questionnaires (during COVID infection and at follow-up), this self-assessed change was only assessed in the follow-up questionnaire, rather than actively comparing their self-rated health between these two questionnaires.

• How do the authors account for a significant symptom burden in the PCR negative group, have they adjusted/factored for baseline co-morbidities?

• I would concur with the reviewers that BMI should be included.

• The authors should mention the predominant ethnicity of their population.

• How do they account for the deterioration in health not being related to mental health e.g. due to isolation from shielding etc

Discussion

• The authors identify that their method of questioning means that the participants’ present state influences their self-reported change in health and mentions this may be a positive. It may also mean that patients who are experiencing symptoms are over-reporting how well they used to be and can introduce recall bias. This should be mentioned within their discussion as it is a limitation of their data.

• COVID-negative patients are more likely to be male which has been reported in other studies and also explains the demographic differences between the exposure groups. This should be mentioned within the discussion.

• Can the authors suggest why being fit is negatively associated with worsening of health?

• I think the result that even young, non-hospitalised patients are not protected from PASC is important to emphasise as it may support public health measures to contain COVID to prevent morbidity, not just death.

• The authors compare their results to [12]; how many % of their patients had at least one persisting symptom? Again, how do they define disease severity and is that comparable to the reference’s method?

• The authors should also compare their non-hospitalised population to the hospitalised population with PASC e.g. their cohort seems very young which may be because it is a non-hospitalised group.

• The authors should specify exactly which results are comparable to references [6-8]; for example gender was associated with persistent symptoms in reference 6 but is not associated with worsening of health in the authors’ paper.

• There are multiple other limitations in the paper that the authors have not mentioned and should address including

o Participants are invited via text messaging and email and surveys are online; these may exclude groups which do not have access to these/face language barriers/less advantaged groups/elderly etc.

o Participants were overwhelmingly women, how does this affect results?

o Multiple symptoms typically associated with PASC e.g. cough are more predominant within the COVID-negative group at follow-up; why? This is an essential part which needs to be addressed; it appears to me most symptoms are more common with the COVID-negative group.

o Why then is self-reported health worse among the COVID-positive group?

o The authors have not assessed for mental health outcomes

• It would be interesting to see more discussion about income and symptoms

• One strength of the study is that the authors are able to compare COVID-negative and positive patients’ symptoms; the authors should emphasise this.

Reviewer #3: Comments to the Author

Thank you for the opportunity to review this manuscript which addresses an important aspect of the COVID-19 disease. The paper provides useful insight into persisting COVID-19 symptoms in a large cohort of the non-hospitalized patients. It is a helpful contribution to the literature and adds value to the body of knowledge.

General comments:

There are three broad areas of concern; the first is the background. The authors can provide a more elaborate study background/ introduction to justify the aim of the study.

The second is the lack of structure in the methods section. The authors should consider using a structured format (STROBE) with a careful definition of the measures assessed and a detailed data analysis plan that led to the results reported.

Another major issue in the paper's methodology was the absence of a follow-up RT PCR test. The test would help eliminate bias and confirm, especially in participants with negative baseline but with respiratory symptoms during follow-up, that they did not get infected within the 3-8 months between baseline and follow-up. Hence, the symptoms reported at follow-up. Similarly, in cases that had tested positive at baseline, there remains a possibility that they could have been re-infected during the interim period between baseline and follow-up; hence, the symptoms reported again. Current evidence suggests reinfection may occur in immunocompetent individuals shortly after recovery from mild COVID-19. [Lee JS, Kim SY, Kim TS, et al. Evidence of Severe Acute Respiratory Syndrome Coronavirus 2 Reinfection After Recovery from Mild Coronavirus Disease 2019. Clin Infect Dis. 2020 Nov 21:ciaa1421. doi: 10.1093/cid/ciaa1421.]

A repeat RT PCR test would have resolved these, as seen in several studies assessing persistent symptoms of COVID-19. [Carfì A, et al. Persistent Symptoms in Patients After Acute COVID-19. JAMA. 2020;324(6):603-5. AND Tenforde MW, et al. Symptom duration and risk factors for delayed return to usual health among outpatients with COVID-19 in a multistate health care systems network—United States, March–June 2020. Morbidity and Mortality Weekly Report. 2020;69(30):993.] If the repeat RT PCR was not done for both groups at follow-up, this should be listed as a limitation in the discussion section.

The third concern is the repetitive texts seen across the result and discussion sections. The authors can elaborate on the pathophysiology of persistent COVID-19 symptoms and how it might explain the symptoms reported by participants with positive baseline tests. Also, the authors should correct the incomplete citation of references 3,5 and 12.

Specific comments are stated below:

Introduction:

1. The authors should provide a more elaborate background to the study aim. For example, what is the burden of COVID-19 disease in the population studied (Norway)? What does the current literature on the pathophysiology of COVID-19 suggest? What gap in literature creates a justification for the present study? A sentence or two on the potential implication of the findings from the study aim?

2. Please take special note that acronyms should be defined the first time they are used in the manuscript. For example, COVID-19, SARS-CoV-2, and RT PCR.

Methodology:

1. The authors should consider providing structure to the methods section, perhaps using the Strengthening the Reporting of Observational Studies in Epidemiology (STROBE) framework as a guide [Vandenbroucke JP, et al. STROBE Initiative. Strengthening the Reporting of Observational Studies in Epidemiology (STROBE): explanation and elaboration. PLoS Med. 2007 Oct 16;4(10):e297.]

For example, explicit statements on the following sections with sub-headers, in the following order: Study design and study population; Ethics approval; Measures and instruments; and Statistical analysis.

2. Page 4, line 8: The authors should define the measure “fitness” in the methodology?

3. Page 4, line 12: The authors should state the criteria used to stratify or categorize “COVID-19 severity of disease”? And define the different categories. For example, what constituted a mild or severe disease in the current study.

4. Were symptoms present even before the COVID-19 infection? This is an important information since some patients might have confounding comorbidities. The authors should specify if this information was taken into account and possibly elaborate in the Discussion section.

5. For those who reported persistent symptoms, was there any procedure to rule out any illness asides from COVID-19 disease? Or by reporting seemingly persistent COVID-19 like symptoms, was it implicitly assumed that the symptom must be linked to the former COVID-19 disease in the positive patients at baseline. If no measure was taken, this may be stated as part of the study limitations.

6. Were participants tested for SARS-CoV-2 infection again before the follow-up questionnaire was administered to confirm “persistent symptoms”? Several studies have reported cases of reinfection in immunocompetent individuals shortly after recovery from mild COVID-19. [Example: Lee JS, Kim SY, Kim TS, et al. Evidence of Severe Acute Respiratory Syndrome Coronavirus 2 Reinfection After Recovery from Mild Coronavirus Disease 2019. Clin Infect Dis. 2020 Nov 21:ciaa1421. doi: 10.1093/cid/ciaa1421.] If participants were not tested at follow-up, this should also be listed as a limitation.

7. The authors should state if therapy (medications etc.) taken by participants that were positive at baseline were considered when collecting data as it is a possible confounding factor.

Results:

1. The sentence “A total of 8786 participants were included.” can be rephrased to “A total of 8786 participants were enrolled in the study”.

2. Table 1: The proportion of females who were negative for SARS-CoV-2 and complete the questionnaire at 3 to 8 months was stated as 76%. The proportion should be 75% (4520/6006 X 100). Kindly correct.

3. Table 2 can be better organized with OR (95% CI) placed in a single column.

4. How were the variables “Fitness- bad shape, fairly fit, fit” AND “Household income- <300k, 300-600k, 600-1000k, and >1000k” analyzed? The authors can provide more clarity by stating the reference as done previously. For example, was the reference "bad shape" for Fitness and "<300k" for NOK?

5. Table 2: The variable Fitness has a p-value <0.001, however, the OR- 2.0 is not within the 95% CI range 0.44-0.57. Can the authors clarify?

6. Table 3: The statement “The average follow-up time was 132 days (SD=35 days) after testing (Table 3)” is not indicated in table 3.

7. Table 3 demonstrates a significant percentage of participants negative at baseline now presenting with COVID-like symptoms (cough, sore throat, body ache, nasal symptoms and headache) and a higher proportion of the negative participants presenting symptomatically compared to positive participants at baseline. The challenge with interpreting these findings is the absence of follow-up SARS-CoV-2 tests on both groups. This is because it is possible that some of the patients who had been negative at baseline over the 3-8 months of follow-up were infected and might have self-isolated and not reported for testing at the study centres. Hence, the symptoms subsequently observed during the follow-up review.

So, we cannot with all certainty infer that participants with +ve RT PCR at baseline had higher odds of “persistent symptoms” than the population assessed who were -ve at baseline assessment. Instead, we can infer that they perceived a significant change in their health status based on the measure "Self-assessed change in health last 1 year (summer)".

8. Page 11, line 3: The authors should correct the "gender (p=0,8)" to "Neither gender (p=0.8)".

9. Table 4: What was the reference value for the “Household income”? The authors should include this as part of the footnote labelled “d”

10. Page 12, line 18-22. There is some repetition of the results, specifically with the text supporting Figure 3.

Discussion

1. Page 13, line 11-12: The statement “where healthcare workers and patients with risk factors for severe disease were prioritized and therefore overrepresented in the negative group” should be revised. While healthcare workers were overrepresented in the negative group, the same does not apply to participants with risk factors for severe diseases. For example, in the negative group, only 39% had chronic diseases, and 61% did not; similarly, about 44% were smokers, and 56% were not.

2. The authors could include a paragraph discussing the pathophysiological approach of the subject of persistent Covid-19 symptoms, referring to the hypothesis proposed in the literature, such as inflammatory post-infectious syndrome, vasculitis, endothelial dysfunction, and social isolation. The authors can refer to the articles:

Sollini M, et al. Vasculitis changes in COVID-19 survivors with persistent symptoms: an [18F] FDG-PET/CT study. Eur J Nucl Med Mol Imaging. 2020. OR Libby P, Lüscher T. COVID-19 is, in the end, an endothelial disease. Eur Heart J. 2020;41:3038–44. OR Garg P, et al. The "post-COVID" syndrome: how deep is the damage? J Med Virol. 2020:1–2.

References

1. Ref 3: Correct to- Rogers JP, Chesney E, Oliver D, Pollak TA, McGuire P, Fusar-Poli P, et al. Psychiatric and neuropsychiatric presentations associated with severe coronavirus infections: a systematic review and meta-analysis with comparison to the COVID-19 pandemic. Lancet Psychiatry. 2020 Jul;7(7):611-627. doi: 10.1016/S2215-0366(20)30203-0.

2. Ref 5: Correct to- Troyer EA, Kohn JN, Hong S. Are we facing a crashing wave of neuropsychiatric sequelae of COVID-19? Neuropsychiatric symptoms and potential immunologic mechanisms. Brain Behav Immun. 2020 Jul;87:34-39. doi: 10.1016/j.bbi.2020.04.027.

3. Ref 12: Correct to- Huang C, Huang L, Wang Y, Li X, Ren L, Gu X, et al. 6-month consequences of COVID-19 in patients discharged from hospital: a cohort study. Lancet. 2021 Jan 16;397(10270):220-232. doi: 10.1016/S0140-6736(20)32656-8.

4. References 6 and 7 can be replaced with peer-reviewed articles.

6. PLOS authors have the option to publish the peer review history of their article (what does this mean?). If published, this will include your full peer review and any attached files.

Reviewer #1: No

Reviewer #2: No

Reviewer #3: No

---

## [Author Response · Author response to Decision Letter 0]

4 Jul 2021

Responses to Editor and Reviewers

These are also enclosed in separate letters. 

PONE-D-21-09363

Persisting Symptoms Three to Eight Months after Non-Hospitalized COVID-19, a Prospective Cohort Study in 8786 participants

PLOS ONE

Additional Editor Comments:

See above.

Journal Requirements:

This has been done. 

2. Please include additional information regarding the survey or questionnaire used in the study and ensure that you have provided sufficient details that others could replicate the analyses. For instance, if you developed a questionnaire as part of this study and it is not under a copyright more restrictive than CC-BY, please include a copy, in both the original language and English, as Supporting Information. If the original language is written in non-Latin characters, for example Amharic, Chinese, or Korean, please use a file format that ensures these characters are visible.

The translated questionnaires have been included in Supplement 2

3. Please state whether you validated the questionnaire prior to testing on study participants. Please provide details regarding the validation group within the methods section.

The endpoint was a question from a validated questionnaire whereas the other questions were not validated. This has been stated in Supplement 2. 

4. Thank you for providing the following Funding Statement: 

"AS and KTK have worked on the project paid from the company Age Labs.

The funders had no role in study design, data collection and analysis, decision to

publish, or preparation of the manuscript."

We note that one or more of the authors is affiliated with the funding organization, indicating the funder may have had some role in the design, data collection, analysis or preparation of your manuscript for publication; in other words, the funder played an indirect role through the participation of the co-authors.

If the funding organization did not play a role in the study design, data collection and analysis, decision to publish, or preparation of the manuscript and only provided financial support in the form of authors' salaries and/or research materials, please review your statements relating to the author contributions, and ensure you have specifically and accurately indicated the role(s) that these authors had in your study in the Author Contributions section of the online submission form. Please make any necessary amendments directly within this section of the online submission form. Please also update your Funding Statement to include the following statement: “The funder provided support in the form of salaries for authors [insert relevant initials], but did not have any additional role in the study design, data collection and analysis, decision to publish, or preparation of the manuscript. The specific roles of these authors are articulated in the ‘author contributions’ section.”

We have updated the funding statement. The updated statement together with a competing interest statement have been uploaded together in a word file. 

If the funding organization did have an additional role, please state and explain that role within your Funding Statement.

It did not. 

Please also provide an updated Competing Interests Statement declaring this commercial affiliation along with any other relevant declarations relating to employment, consultancy, patents, products in development, or marketed products, etc. 

This has been done. 

Data, even deidentified data contains sensitive patient information and the data can only be shared after an approval process involving the Ethics committee and the Data Protection Officer at our hospital. Both will follow the EU-GDPR rules. 

The Data Access Committee of the Norwegian Corona Cohort must be contacted through the PI:

Arne Søraas

Department of Microbiology, 

Oslo University Hospital, 

Rikshospitalet, NO-0372 Oslo, 

Norway 

 

Reviewers' comments:

Reviewer's Responses to Questions

Comments to the Author

1. Is the manuscript technically sound, and do the data support the conclusions?

Reviewer #1: Yes

Reviewer #2: Yes

Reviewer #3: Partly

2. Has the statistical analysis been performed appropriately and rigorously? 

Reviewer #1: Yes

Reviewer #2: Yes

Reviewer #3: Yes

3. Have the authors made all data underlying the findings in their manuscript fully available?

Reviewer #1: No

Reviewer #2: No

Reviewer #3: Yes

The data underlying the manuscript is personally identifiable data under the EU-GDPR regulations making publicly sharing of the data extremely difficult. 

4. Is the manuscript presented in an intelligible fashion and written in standard English?

Reviewer #1: Yes

Reviewer #2: Yes

Reviewer #3: No

5. Review Comments to the Author

Reviewer #1: I suggest to review recent literature data on the topic

e.g.

Sudre CH, Murray B, Varsavsky T, Graham MS, Penfold RS, Bowyer RC, et al. Attributes and predictors of Long-COVID: analysis of COVID cases and their symptoms collected by the Covid Symptoms Study App. medRxiv. Cold Spring Harbor Laboratory Press; 2020;2020.10.19.20214494. and perform similar analysis to confirm results

The manuscripts main findings is that the participants report a deterioration of their health after COVID-19. The symptoms was mainly reported in July 2020 before long-COVID was to our knowledge reported in the Norwegian media. As our finding of a deterioration of health was not expected we did not collect the range of explanatory data that Sudre et al collected and cannot report those. However, the data we did collect included symptoms directly related to COVID-19 and those did not explain the worsening of health reported by our participants. 

Reviewer #2: Review: Persisting Symptoms Three to Eight Months after Non-Hospitalized COVID-19, a Prospective Cohort Study in 8786 participants

• Overall, I think that the question (if non-hospitalised adults continue to experience symptoms from PASC) is important and not fully answered in current literature

• The authors’ methodology to answer this question is to use a survey to compare self-reported health outcomes between COVID positive and negative patients at an adequate time post-infection; the use of a control group is an important positive

• However, the discussion is very limited and does not fully explain the research findings

The discussion has been extended. 

Abstract

• I think it would be important to mention the numbers of COVID-positive and negative patients within the abstract, given most of the 8786 participants are COVID-negative

This has been mentioned.

• While the abstract mentions the most important conclusion, it does not mention the other important data e.g. important symptoms such as cough are more common within the COVID-negative group

We have now mentioned two of the airway symptoms that were more commonly reported by the control group in the abstract. 

Introduction

• The introduction is limited and the authors have not discussed the current literature available regarding post covid symptoms following hospitalization

The introduction has been expanded. 

• Having 8 references for the first sentence seems excessive, given the first five are not further referenced later

The references has been updated. 

• Saying that ‘outpatients with COVID-19 experience relatively benign symptoms’ may trivialise what are often very debilitating symptoms and the authors should rephrase this.

This has been rephrased and this point has also been included in the discussion where we question whether COVID-19 can be a “mild” disease. 

• Why have the authors picked 3-8 months – was this an a priori decision?

The results are based on the responses to the follow-up questionnaire. This was distributed first three months after disease, and reminders were sent repeatedly approximately every month until a high number of participants had responded. Most participant did responded relatively early (mean 133 days for controls). As long-COVID is a new syndrome any follow-up length is of interest to describe when and which symptoms are present at each timepoint. 

• The introduction should clarify that it is the non-hospitalised COVID-19 patient group which is being assessed

This has now been clarified. 

Materials and methods

• Overall, these are described well

• I think the authors should clarify clearly how they classified severity of disease within a non-hospitalised cohort and the evidence used for this classification. I have normally seen ‘mild COVID’ used to describe non-hospitalised COVID.

This definition was included in the results and has now been moved to the methods section. 

• It would be useful to understand why subjects underwent a swab e.g. contact with COVID positive person, self referral with minimum symptoms, routine referral with no symptoms, ie were the negative patients likely to have negative swabs

In Norway, testing for SARS-CoV-2 is available free of cost and at the time of inclusion for the study, testing was almost exclusively conducted on symptomatic cases. Testing was usually done after self-referral because of symptoms. We have included this information in the Materials and Methods section. 

Results

• Overall the results can be understood, although some of the tables could be simplified

The tables have been simplified. 

• In the baseline questionnaire, COVID-negative patients were more likely to experience sore throat than COVID-positive patients. As this is the only symptom they were more likely to experience, I think it should be mentioned within the results section

It is correct COVID-negative group reported more airway symptoms at follow-up. This has now been fully presented in the results section. This is an important point that is also included in the discussion. 

• Can the authors add a sentence about household income to clarify if it is increased or decreased household income which is associated with a worsening of health?

Thank you for this input. The direction of association has been included in this sentence. For brevity, the odds ratios have not been included both household income and fitness were multilevel variables. 

• The most important finding that the authors have focussed on is ‘self-assessed change in health in last one year’. I think it is worth clarifying that while the authors administered two questionnaires (during COVID infection and at follow-up), this self-assessed change was only assessed in the follow-up questionnaire, rather than actively comparing their self-rated health between these two questionnaires.

This point has been added to this section in the Results. 

• How do the authors account for a significant symptom burden in the PCR negative group, have they adjusted/factored for baseline co-morbidities?

This is a part of the discussion. It is likely that individuals either reporting or having more airway symptoms were more likely to be tested for COVID-19 as symptoms was a prerequisite for being tested. Thus, the COVID-negative group can be enriched with individuals often exhibiting such symptoms. Importantly, this increased symptom burden do seem to negatively influence their self-reported health compared to one year ago as 18% repored a worsening while the Norwegian norm for this questionnaire item (when dichotomized) is only 12%. 

• I would concur with the reviewers that BMI should be included.

We have not been able to access these data. Although they have been collected from most of the patients, they have been protected because of their personally identifiable nature and are not available for analysis at this point. 

• The authors should mention the predominant ethnicity of their population.

A sentence on ethnicity has been added to the Results section. 97% of respondents to this questionnaire item reported being Caucasian. 

• How do they account for the deterioration in health not being related to mental health e.g. due to isolation from shielding etc

The deterioration of health may be caused by mental health, and the study does not address this directly. COVID has even been reported to cause mental health problems. However, the period of isolation after COVID were less than a two weeks and we find it unlikely that the period of isolation would influence the self-reported health after three months and up to eight months. 

Discussion

• The authors identify that their method of questioning means that the participants’ present state influences their self-reported change in health and mentions this may be a positive. It may also mean that patients who are experiencing symptoms are over-reporting how well they used to be and can introduce recall bias. This should be mentioned within their discussion as it is a limitation of their data.

The possibility of recall bias is included in the Limitation section. The influence of current health on the single-item health transition question that we used are discussed in the beginning of the discussion (…”it has been shown that current health-related quality of life heavily influences the answers”)

• COVID-negative patients are more likely to be male which has been reported in other studies and also explains the demographic differences between the exposure groups. This should be mentioned within the discussion.

The COVID-positive, patients were more likely to be male as has also been reported. This has been added to the discussion. 

• Can the authors suggest why being fit is negatively associated with worsening of health?

We have added a sentence suggesting that this may represent a reversion to the mean, but this is speculative. 

• I think the result that even young, non-hospitalised patients are not protected from PASC is important to emphasise as it may support public health measures to contain COVID to prevent morbidity, not just death.

Yes. We have added the point both to the introduction and the discussion and conclusion. 

• The authors compare their results to [12]; how many % of their patients had at least one persisting symptom? Again, how do they define disease severity and is that comparable to the reference’s method?

The key finding is the self-reported worsening of health. We did not ask about other persisting symptoms except those directly associated with COVID-19 like changes to the sense of smell and taste, cough and fever and reports those. We did stratified analyses based on severity now defined in the last part of the methods section. The reference has been changed. 

• The authors should also compare their non-hospitalised population to the hospitalised population with PASC e.g. their cohort seems very young which may be because it is a non-hospitalised group.

Hospitalized patients are not part of the manuscript because they were very few. 

• The authors should specify exactly which results are comparable to references [6-8]; for example gender was associated with persistent symptoms in reference 6 but is not associated with worsening of health in the authors’ paper.

This sentence has been changed. 

• There are multiple other limitations in the paper that the authors have not mentioned and should address including

o Participants are invited via text messaging and email and surveys are online; these may exclude groups which do not have access to these/face language barriers/less advantaged groups/elderly etc.

This has been added as a limitation.

o Participants were overwhelmingly women, how does this affect results?

The odds ratio in stratified analysis in the adjusted model was 3.2 for women and 2.4 for men. This has not been included in the manuscript for brevity. 

o Multiple symptoms typically associated with PASC e.g. cough are more predominant within the COVID-negative group at follow-up; why? This is an essential part which needs to be addressed; it appears to me most symptoms are more common with the COVID-negative group.

This is included in the discussion and the sentence has been rewritten: “Such symptoms were a prerequisite for being tested, and individuals frequently experiencing such symptoms are therefore expected to be tested more often even if they did not have COVID-19 and therefore the negative group will be enriched with such individuals.”

o Why then is self-reported health worse among the COVID-positive group?

This is a key open question left after our study which the literature is now beginning to answer. Our follow-up questionnaire had an open text field where participants were asked to report how they felt. 

We have sought answers to your question here (not reported in this manuscript) and often, cognitive symptoms (but not so much depression) are reported by those also reporting a worsening of health. These are qualitative data that are not fully analyzed and are therefore not reported. 

o The authors have not assessed for mental health outcomes

True.

• It would be interesting to see more discussion about income and symptoms

Household income were only weakly associated with symptoms as shown in the updated Table 4. 

• One strength of the study is that the authors are able to compare COVID-negative and positive patients’ symptoms; the authors should emphasise this.

This has been emphasized. 

Reviewer #3: Comments to the Author

Thank you for the opportunity to review this manuscript which addresses an important aspect of the COVID-19 disease. The paper provides useful insight into persisting COVID-19 symptoms in a large cohort of the non-hospitalized patients. It is a helpful contribution to the literature and adds value to the body of knowledge.

General comments:

There are three broad areas of concern; the first is the background. The authors can provide a more elaborate study background/ introduction to justify the aim of the study.

The introduction has been expanded significantly. 

The second is the lack of structure in the methods section. The authors should consider using a structured format (STROBE) with a careful definition of the measures assessed and a detailed data analysis plan that led to the results reported.

We have followed the STROBE guidelines when we wrote the report. 

Another major issue in the paper's methodology was the absence of a follow-up RT PCR test. The test would help eliminate bias and confirm, especially in participants with negative baseline but with respiratory symptoms during follow-up, that they did not get infected within the 3-8 months between baseline and follow-up. Hence, the symptoms reported at follow-up. Similarly, in cases that had tested positive at baseline, there remains a possibility that they could have been re-infected during the interim period between baseline and follow-up; hence, the symptoms reported again. Current evidence suggests reinfection may occur in immunocompetent individuals shortly after recovery from mild COVID-19. [Lee JS, Kim SY, Kim TS, et al. Evidence of Severe Acute Respiratory Syndrome Coronavirus 2 Reinfection After Recovery from Mild Coronavirus Disease 2019. Clin Infect Dis. 2020 Nov 21:ciaa1421. doi: 10.1093/cid/ciaa1421.]

A follow-up RT PCR test was not performed, but testing is free in Norway and everyone with symptoms are encouraged to report immediately for testing. Therefore most incident cases between baseline and follow-up would be diagnosed and reported to the Norwegian mandatory system infectious diseases from where our study has recently obtained data. The data we have obtained from this system shows that 30 participants of the negative group had a positive SARS-CoV-2 RT PCR test in the time interval from baseline to completing follow-up. In this period, we had very little COVID in Norway so the incidence 30/7992 (=0.4%) is plausible. They were too few to significantly influence the incidence of airway symptoms in the COVID-negative group at follow-up.

The major finding of the study is the worsening of self-reported health compared to one year ago among the COVID-positive group and COVID cases in the control group will serve to reduce the observed difference between the positive and negative groups. 

Some of the COVID-19 positive participants may have been reinfected, but that would probably be an even lower number than among the controls and therefore too few to impact the result. 

A repeat RT PCR test would have resolved these, as seen in several studies assessing persistent symptoms of COVID-19. [Carfì A, et al. Persistent Symptoms in Patients After Acute COVID-19. JAMA. 2020;324(6):603-5. AND Tenforde MW, et al. Symptom duration and risk factors for delayed return to usual health among outpatients with COVID-19 in a multistate health care systems network—United States, March–June 2020. Morbidity and Mortality Weekly Report. 2020;69(30):993.] If the repeat RT PCR was not done for both groups at follow-up, this should be listed as a limitation in the discussion section.

This has been added as a limitation. 

The third concern is the repetitive texts seen across the result and discussion sections. The authors can elaborate on the pathophysiology of persistent COVID-19 symptoms and how it might explain the symptoms reported by participants with positive baseline tests. Also, the authors should correct the incomplete citation of references 3,5 and 12.

The text has been thoroughly changed to remove repetitive text, the pathophysiology has been elaborated and all references updated. 

Specific comments are stated below:

Introduction:

1. The authors should provide a more elaborate background to the study aim. For example, what is the burden of COVID-19 disease in the population studied (Norway)? What does the current literature on the pathophysiology of COVID-19 suggest? What gap in literature creates a justification for the present study? A sentence or two on the potential implication of the findings from the study aim?

The introduction has been completely rewritten to accommodate these suggestions. 

2. Please take special note that acronyms should be defined the first time they are used in the manuscript. For example, COVID-19, SARS-CoV-2, and RT PCR.

Methodology:

1. The authors should consider providing structure to the methods section, perhaps using the Strengthening the Reporting of Observational Studies in Epidemiology (STROBE) framework as a guide [Vandenbroucke JP, et al. STROBE Initiative. Strengthening the Reporting of Observational Studies in Epidemiology (STROBE): explanation and elaboration. PLoS Med. 2007 Oct 16;4(10):e297.]

For example, explicit statements on the following sections with sub-headers, in the following order: Study design and study population; Ethics approval; Measures and instruments; and Statistical analysis.

The STROBE checklist was followed when writing the manuscript, but some of the sub-headers have been merged for improved readability. 

2. Page 4, line 8: The authors should define the measure “fitness” in the methodology?

This has been added in the Multivariate analysis and confounding section. 

3. Page 4, line 12: The authors should state the criteria used to stratify or categorize “COVID-19 severity of disease”? And define the different categories. For example, what constituted a mild or severe disease in the current study.

This has been defined. 

4. Were symptoms present even before the COVID-19 infection? This is an important information since some patients might have confounding comorbidities. The authors should specify if this information was taken into account and possibly elaborate in the Discussion section.

We have no information about symptoms before COVID-19 infection. This has been specified in the limitations. 

5. For those who reported persistent symptoms, was there any procedure to rule out any illness asides from COVID-19 disease? Or by reporting seemingly persistent COVID-19 like symptoms, was it implicitly assumed that the symptom must be linked to the former COVID-19 disease in the positive patients at baseline. If no measure was taken, this may be stated as part of the study limitations.

This has been included as a limitation. 

6. Were participants tested for SARS-CoV-2 infection again before the follow-up questionnaire was administered to confirm “persistent symptoms”? Several studies have reported cases of reinfection in immunocompetent individuals shortly after recovery from mild COVID-19. [Example: Lee JS, Kim SY, Kim TS, et al. Evidence of Severe Acute Respiratory Syndrome Coronavirus 2 Reinfection After Recovery from Mild Coronavirus Disease 2019. Clin Infect Dis. 2020 Nov 21:ciaa1421. doi: 10.1093/cid/ciaa1421.] If participants were not tested at follow-up, this should also be listed as a limitation.

This has been included as a limitation.

7. The authors should state if therapy (medications etc.) taken by participants that were positive at baseline were considered when collecting data as it is a possible confounding factor.

It is possible that differential medication use not accounted for between the groups is a confounding factor. However, no medication except antipyretics were regularly used for non-hospitalized COVID-19 in Norway at that time, making the impact of such differences between the groups small. The possibility of unmeasured confounding is included in the beginning under Limitations. 

Results:

1. The sentence “A total of 8786 participants were included.” can be rephrased to “A total of 8786 participants were enrolled in the study”.

This section has been rewritten. 

2. Table 1: The proportion of females who were negative for SARS-CoV-2 and complete the questionnaire at 3 to 8 months was stated as 76%. The proportion should be 75% (4520/6006 X 100). Kindly correct.

This is caused by missing information on gender from a small number of participants. The number of missing values on each variable has now been reported in Table 1. The variable household income has a high number of missing values as it was not a part of the baseline form although it is a baseline variable. It was a part of the follow-up form instead. 

3. Table 2 can be better organized with OR (95% CI) placed in a single column.

We have removed odds ratios to simplify the tables after consideration in our group and after the comment from reviewer 2. 

4. How were the variables “Fitness- bad shape, fairly fit, fit” AND “Household income- <300k, 300-600k, 600-1000k, and >1000k” analyzed? The authors can provide more clarity by stating the reference as done previously. For example, was the reference "bad shape" for Fitness and "<300k" for NOK?

This has now been done the same way as for the other variables. 

5. Table 2: The variable Fitness has a p-value <0.001, however, the OR- 2.0 is not within the 95% CI range 0.44-0.57. Can the authors clarify?

The Fitness variable has now been presented the same way as the other variables. The OR 2.0 was the inverse of the presented 95% CI and this is not correct. The fitness variable as it is provided now is both more detailed and correct. A Self-reported fitness was strongly associated with SARS-CoV-2 positivity. (We speculate that this may be because many of the early cases in Norway were ski tourists returning from the Austrian and Italian Alps and ski tourists tend to be relatively fit compared to the rest of the population). 

6. Table 3: The statement “The average follow-up time was 132 days (SD=35 days) after testing (Table 3)” is not indicated in table 3.

This reference has been removed. The follow-up time was 133 days for the controls and 125 days for the SARS-CoV-2 positive cases. 

7. Table 3 demonstrates a significant percentage of participants negative at baseline now presenting with COVID-like symptoms (cough, sore throat, body ache, nasal symptoms and headache) and a higher proportion of the negative participants presenting symptomatically compared to positive participants at baseline. The challenge with interpreting these findings is the absence of follow-up SARS-CoV-2 tests on both groups. This is because it is possible that some of the patients who had been negative at baseline over the 3-8 months of follow-up were infected and might have self-isolated and not reported for testing at the study centres. Hence, the symptoms subsequently observed during the follow-up review.

As the COVID-19 was extremely rare during the summer of 2020 in Norway it is highly unlikely that the higher symptom burden among the negative group is caused by COVID-19. See also under general comments above. 

So, we cannot with all certainty infer that participants with +ve RT PCR at baseline had higher odds of “persistent symptoms” than the population assessed who were -ve at baseline assessment. Instead, we can infer that they perceived a significant change in their health status based on the measure "Self-assessed change in health last 1 year (summer)".

Yes. That is the our conclusion. We do not propose that COVID-19 protects against airway symptoms during the months after disease. This point is discussed under reviewer 2, results, bullet point 5: 

“This is a part of the discussion. It is likely that individuals either reporting or having more airway symptoms were more likely to be tested for COVID-19 as symptoms was a prerequisite for being tested. Thus, the COVID-negative group can be enriched with individuals often exhibiting such symptoms. Importantly, this increased symptom burden do seem to negatively influence their self-reported health compared to one year ago as 18% reported a worsening while the Norwegian norm for this questionnaire item (when dichotomized) is only 12%.”

8. Page 11, line 3: The authors should correct the "gender (p=0,8)" to "Neither gender (p=0.8)".

This section has been completely removed because these numbers are already presented in Table 4.

9. Table 4: What was the reference value for the “Household income”? The authors should include this as part of the footnote labelled “d”

The odds ratio have been removed. 

10. Page 12, line 18-22. There is some repetition of the results, specifically with the text supporting Figure 3.

Some of the text is a repetition to ensure that the figure with its text can be read without reading the rest of the text. 

Discussion

1. Page 13, line 11-12: The statement “where healthcare workers and patients with risk factors for severe disease were prioritized and therefore overrepresented in the negative group” should be revised. While healthcare workers were overrepresented in the negative group, the same does not apply to participants with risk factors for severe diseases. For example, in the negative group, only 39% had chronic diseases, and 61% did not; similarly, about 44% were smokers, and 56% were not.

This sentence has been changed. Thank you for pointing this out. 

2. The authors could include a paragraph discussing the pathophysiological approach of the subject of persistent Covid-19 symptoms, referring to the hypothesis proposed in the literature, such as inflammatory post-infectious syndrome, vasculitis, endothelial dysfunction, and social isolation. The authors can refer to the articles:

Sollini M, et al. Vasculitis changes in COVID-19 survivors with persistent symptoms: an [18F] FDG-PET/CT study. Eur J Nucl Med Mol Imaging. 2020. OR Libby P, Lüscher T. COVID-19 is, in the end, an endothelial disease. Eur Heart J. 2020;41:3038–44. OR Garg P, et al. The "post-COVID" syndrome: how deep is the damage? J Med Virol. 2020:1–2.

This has been done. 

References

1. Ref 3: Correct to- Rogers JP, Chesney E, Oliver D, Pollak TA, McGuire P, Fusar-Poli P, et al. Psychiatric and neuropsychiatric presentations associated with severe coronavirus infections: a systematic review and meta-analysis with comparison to the COVID-19 pandemic. Lancet Psychiatry. 2020 Jul;7(7):611-627. doi: 10.1016/S2215-0366(20)30203-0.

2. Ref 5: Correct to- Troyer EA, Kohn JN, Hong S. Are we facing a crashing wave of neuropsychiatric sequelae of COVID-19? Neuropsychiatric symptoms and potential immunologic mechanisms. Brain Behav Immun. 2020 Jul;87:34-39. doi: 10.1016/j.bbi.2020.04.027.

3. Ref 12: Correct to- Huang C, Huang L, Wang Y, Li X, Ren L, Gu X, et al. 6-month consequences of COVID-19 in patients discharged from hospital: a cohort study. Lancet. 2021 Jan 16;397(10270):220-232. doi: 10.1016/S0140-6736(20)32656-8.

4. References 6 and 7 can be replaced with peer-reviewed articles.

---

## [Decision Letter · Decision Letter 1]

2 Aug 2021

Persisting Symptoms Three to Eight Months after Non-Hospitalized COVID-19, a Prospective Cohort Study

PONE-D-21-09363R1

Dear Dr. Søraas,

We’re pleased to inform you that your manuscript has been judged scientifically suitable for publication and will be formally accepted for publication once it meets all outstanding technical requirements.

Kind regards,

Dong-Yan Jin

Academic Editor

PLOS ONE

Additional Editor Comments (optional):

Reviewers' comments:

Reviewer's Responses to Questions

**Comments to the Author**

1. If the authors have adequately addressed your comments raised in a previous round of review and you feel that this manuscript is now acceptable for publication, you may indicate that here to bypass the “Comments to the Author” section, enter your conflict of interest statement in the “Confidential to Editor” section, and submit your "Accept" recommendation.

Reviewer #3: All comments have been addressed

2. Is the manuscript technically sound, and do the data support the conclusions?

Reviewer #3: Yes

3. Has the statistical analysis been performed appropriately and rigorously? 

Reviewer #3: Yes

4. Have the authors made all data underlying the findings in their manuscript fully available?

Reviewer #3: Yes

5. Is the manuscript presented in an intelligible fashion and written in standard English?

Reviewer #3: Yes

6. Review Comments to the Author

Reviewer #3: (No Response)

7. PLOS authors have the option to publish the peer review history of their article (what does this mean?). If published, this will include your full peer review and any attached files.

Reviewer #3: No

---

## [Editor Report · Acceptance letter]

18 Aug 2021

PONE-D-21-09363R1 

Persisting Symptoms Three to Eight Months after Non-Hospitalized COVID-19, a Prospective Cohort Study 

Dear Dr. Søraas:

I'm pleased to inform you that your manuscript has been deemed suitable for publication in PLOS ONE. Congratulations! Your manuscript is now with our production department. 

Kind regards, 

on behalf of

Professor Dong-Yan Jin 

Academic Editor

PLOS ONE